# Epigenetic analysis of Paget's disease of bone identifies differentially methylated loci that predict disease status

Ilhame Diboun[1], Sachin Wani[2], Stuart H Ralston[2], Omar ME Albagha[1,2]*

[1]Division of Genomic and Translational Biomedicine, College of Health and Life Sciences, Hamad Bin Khalifa University, Doha, Qatar; [2]Centre for Genomic and Experimental Medicine, MRC Institute of Genetics and Molecular Medicine, University of Edinburgh, Edinburgh, United Kingdom

**Abstract** Paget's disease of bone (PDB) is characterized by focal increases in disorganized bone remodeling. This study aims to characterize PDB-associated changes in DNA methylation profiles in patients' blood. Meta-analysis of data from the discovery and cross-validation set, each comprising 116 PDB cases and 130 controls, revealed significant differences in DNA methylation at 14 CpG sites, 4 CpG islands, and 6 gene-body regions. These loci, including two characterized as functional through expression quantitative trait-methylation analysis, were associated with functions related to osteoclast differentiation, mechanical loading, immune function, and viral infection. A multivariate classifier based on discovery samples was found to discriminate PDB cases and controls from the cross-validation with a sensitivity of 0.84, specificity of 0.81, and an area under curve of 92.8%. In conclusion, this study has shown for the first time that epigenetic factors contribute to the pathogenesis of PDB and may offer diagnostic markers for prediction of the disease.

## Introduction

Paget's disease of bone (PDB) is characterized by increased but disorganized bone remodeling, which causes affected bones to enlarge, become weak, and deform. The axial skeleton is predominantly involved and commonly affected sites include the skull, spine, and pelvis. Paget's disease is clinically silent until it has reached an advanced stage at which point irreversible damage to the skeleton has occurred (*Tan and Ralston, 2014*). Bisphosphonates are an effective treatment (*Ralston et al., 2019*) and can often improve bone pain but have a limited impact on other clinical outcomes in patients with advanced disease (*Langston et al., 2010*; *Reid et al., 2011*). On a cellular level, PDB is characterized by increased osteoclast activity and biopsies from affected bone lesions exhibit increase in the number and size of osteoclasts.

Genetic factors play an important role in classical PDB and in monogenic PDB-like syndromes (*Gennari et al., 2019*; *Ralston and Albagha, 2014*). Mutations in *SQSTM1* are the most common cause of PDB, but other susceptibility genes and loci have been identified through genome-wide association studies (*Vallet et al., 2015*; *Albagha et al., 2011*; *Albagha et al., 2010*). These include genes that play an important role in osteoclast differentiation such as *CSF1*, *TNFRSF11A*, and *DCSTAMP*. Additionally, an expression quantitative trait locus (eQTL) in *OPTN* is associated with increased susceptibility to PDB (*Obaid et al., 2015*). Functional analysis using mouse models showed that OPTN is a negative regulator of osteoclast differentiation and mice with loss of *OPTN* function develop PDB-like bone lesions with increasing age (*Obaid et al., 2015*; *Wong et al., 2020*).

Environmental factors also play a role, as evidenced by the fact that the disease is focal in nature and its incidence and severity has diminished in recent years (*Corral-Gudino et al., 2013*). Several

*For correspondence:
oalbagha@hbku.edu.qa

**eLife digest** Our skeleton stays healthy through an endless regeneration process, with specialized cells constantly absorbing and creating new bone tissue. Illnesses emerge when this breaking down and rebuilding cycle becomes imbalanced. For instance, in Paget's disease of bone (PDB for short) the skeleton becomes misshapen and fragile, with complications including pain, fractures, neurological problems, hearing loss and even cancer. For most patients however, symptoms are only present at an advanced stage, when irreversible damage to the skeleton has already occurred.

Certain inherited genetic changes play a role in the development of PDB, but lifestyle and environmental factors are also thought to contribute. Indeed, accumulating evidence suggests that diet, pollution and infection may influence how genes involved in bone metabolism are activated. In this process, the environment may trigger chemical marks to be added onto DNA sequences, which ultimately switches specific genes on and off.

To investigate whether the pattern of chemical marks in individuals with PDB may be characteristic, Diboun et al. scanned the genetic information of over 200 PDB patients, and compared it to healthy counterparts. Combining genomic analysis and machine learning revealed several chemical signatures that were remarkably different in the DNA of PDB individuals. These signatures affected sites close to genes involved in bone development, as well as response to mechanical loading and infection. This provides strong evidence that PDB could be, in part, triggered by the environment, as the placement of these marks is highly influenced by external factors.

This research sheds light onto the underlying changes that trigger PDB. Future experiments should explore whether it may be possible to use these genetic changes to identify patients before the onset of irreversible and debilitating damage.

environmental triggers have been suggested including persistent viral infection, repetitive mechanical loading of the skeleton, low dietary calcium intake, environment pollutants, and vitamin D deficiency (*Ralston and Albagha, 2014*).

The possible role of persistent viral infection with measles and distemper has been studied experimentally. For example, expression of the measles virus nucleocapsid protein in osteoclasts was found to trigger PDB-like phenotype in mice (*Kurihara et al., 2011*; *Kurihara et al., 2006*). However, clinical studies that have sought to detect evidence of viral proteins and nucleic acids in humans with PDB have yielded conflicting results (*Ralston et al., 2019*).

Accumulating evidence suggests that environmental and lifestyle factors can influence gene expression and clinical phenotype in various diseases through epigenetic mechanisms such as changes in DNA methylation. To gain insights into the role of epigenetic DNA methylation in PDB, we have conducted genome-wide profiling of DNA methylation in a cohort of 253 PDB patients and 280 controls and evaluated the predictive role of epigenetic markers in differentiating patients with PDB from controls.

## Results

### Characteristics of study cohort

*Table 1* shows descriptive statistics for the study cohort. PDB cases in the discovery set were slightly older and included more males compared to controls, but no difference in age or gender distribution was found in the cross-validation set. The number of patients with *SQSTM1* mutations was similar in the discovery and cross-validation set and accounts for approximately 14% of PDB cases. All controls were negative for *SQSTM1* mutations as shown in *Table 1*.

### Differentially methylated sites

*Figure 1* shows the study design and summary of differential methylation results. After adjusting for all confounders, differential methylation analysis of the discovery set revealed 419 differentially methylated sites (DMS) with false discovery rate (FDR) < 0.05, of which 57 reached statistical

**Table 1.** Descriptive statistics of the study cohort.

|  | Discovery | | Cross-validation | |
| --- | --- | --- | --- | --- |
|  | PDB case | Control | PDB case | Control |
| Number | 116 | 130 | 116 | 130 |
| Age (years), mean ± SD | 72.1 ± 7.5* | 70.0 ± 7.4* | 72.5 ± 8.7 | 72.3 ± 8.2 |
| Male, n (%) | 65 (56.0)* | 48 (36.9)* | 59 (50.9) | 53 (40.8) |
| Female, n (%) | 51 (44.0)* | 82 (63.1)* | 57 (49.1) | 77 (59.2) |
| *SQSTM1* mutation, n (%) | 16 (13.8) | 0 (0) | 17 (14.6) | 0 (0) |

*P<0.05 comparing Paget's disease (PDB) cases to controls.

significance (FDR < 0.05) in the cross-validation set (*Supplementary file 1*). Meta-analysis of the DMS from discovery and cross-validation revealed 14 Bonferroni significant DMS out of a total of 429,156 tested CpG sites (p<1.17×10$^{-7}$; *Table 2*). The direction of effect for all replicated DMS was identical in the discovery and cross-validation set and shows hypermethylation in PDB cases compared to controls. A Manhattan plot of the results is shown in *Figure 2A*, and a quantile–quantile (Q–Q) plot is presented in *Figure 2—figure supplement 1*.

## Differentially methylated regions

Besides analyzing individual sites, our region-based analysis was intended to uncover densely hyper/hypo-methylated regions with unique effects across the genome in PDB as well as identifying instances where the effect from individual sites is moderate, yet accumulatively significant. We tested natural concentrations of sites with independent effects within CpG islands but also gene bodies and promoter regions, justified by the fact that promoter methylation often suppresses transcription whilst that from the gene body often stimulates gene expression (*Figure 1*).

Evaluation of the 25,773 CpG islands on the array revealed 978 differentially methylated regions (DMR) that were significantly differentially methylated (FDR < 0.05) in the discovery set, of which 111 replicated at the same significance level in the cross-validation set (*Supplementary file 2*). Stringent Bonferroni multiple testing correction revealed four islands that remained significant in the discovery and cross-validation, and these were located near *LTB*, *SKIV2L*, *EBF3*, and *CCND1* (*Table 3*).

Gene body analysis revealed 258 (FDR < 0.05) replicated DMR out of a total of 947 differentially methylated genes initially identified in the discovery set (*Supplementary file 3*). Six gene body DMR reached significance after Bonferroni correction in both the discovery and cross-validation set (*Table 3*). In the context of promoter regions, evidence for FDR significant association with the disease was equally observed in the discovery and cross-validation set for 27 promoters DMR (*Supplementary file 4*), but none reached significance after Bonferroni correction. *Figure 2B and C* show a regional plot for DMR within LTB and HSPA13 from island and gene body analysis respectively, highlighting the co-occurrence of multiple, yet independent, differentially methylated sites along each region.

## Mapping common regulatory patterns of DNA methylation into functional networks

To gain further insight into the pathology of PDB, we explored common methylation patterns amongst functional keywords identified as significantly over-represented amongst the *Pooled* sites (a unified list of 2847 candidate CpGs identified from the DMS and DMR analysis, refer to Materials and methods). *Figure 3* shows a graphical representation of these functional keywords. In addition to bone-related cells, there is a strong presence of immune cells linked to key biological processes including proliferation, differentiation, autophagy, and cell death. Furthermore, virus, cytokines, and interferon-gamma were among the over-represented keywords. The process of ubiquitination lies at the center of the graph with the largest number of links in the network.

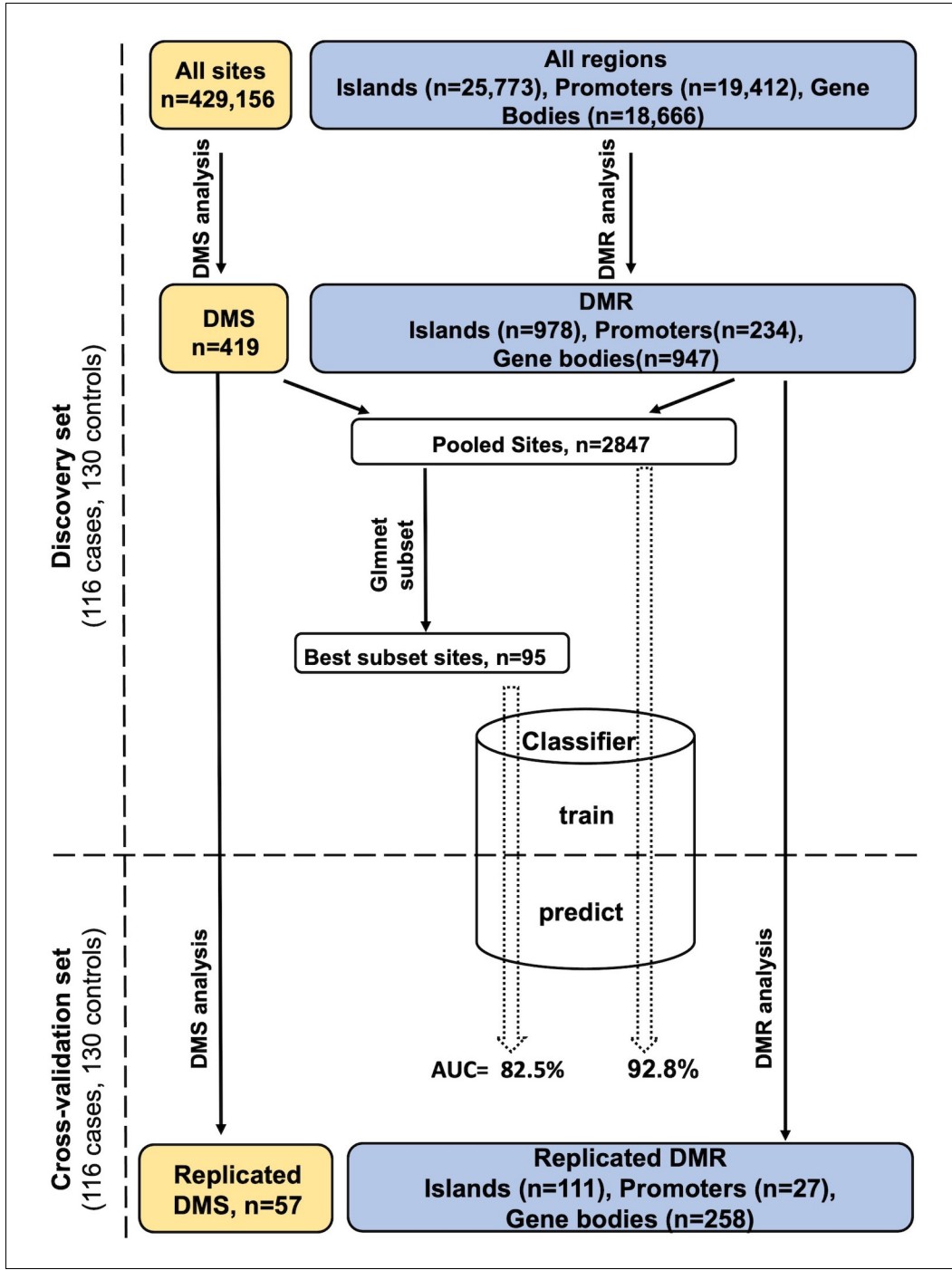

**Figure 1.** Study design and analysis workflow. Differentially methylated sites (DMS) and differentially methylated regions (DMR) were analyzed using, the general/generalized linear model, respectively, in the discovery set. Those reaching FDR < 0.05 were tested in the cross-validation set to identify DMS/DMR that replicate at the same significance level. The DMS and the important sites within DMR were pooled together giving rise to the *Pooled sites* (refer to Materials and methods), of these a best PDB discriminatory subset was obtained using the Lasso and Elastic-Net regression. A multivariate classifier based on the discovery measurement of the Pooled/Best subset sites yielded an AUC value of 92.8% and 82.5%, respectively, when tested in the cross-validation.

**Table 2.** Differentially methylated CpG sites (DMS) in Paget's disease of bone.

| CpG Site | | | Discovery | | Cross-validation | | Meta-analysis | | Annotations |
|---|---|---|---|---|---|---|---|---|---|
| Probe ID | Chr | Position | Δ Beta* | p-value | Δ Beta* | p-value | Δ Beta* | p-value | Nearest gene |
| cg10290814 | 17 | 7284330 | −0.018 | $1.2 \times 10^{-6}$ | −0.015 | $1.4 \times 10^{-4}$ | −0.017 | $2.3 \times 10^{-10}$ | TNK1 |
| cg19361865 | 1 | 220922163 | −0.014 | $5.4 \times 10^{-6}$ | −0.012 | $9.7 \times 10^{-5}$ | −0.013 | $7.6 \times 10^{-10}$ | MOSC2 |
| cg09152582 | 1 | 88928362 | −0.021 | $2.1 \times 10^{-5}$ | −0.018 | $3.5 \times 10^{-5}$ | −0.019 | $1.1 \times 10^{-9}$ | PKN2-AS1 |
| cg09260089 | 10 | 134599860 | −0.024 | $4.6 \times 10^{-5}$ | −0.024 | $1.2 \times 10^{-4}$ | −0.024 | $9.5 \times 10^{-9}$ | NKX6-2 |
| cg24879273 | 10 | 102989645 | −0.026 | $4.9 \times 10^{-5}$ | −0.016 | $1.7 \times 10^{-4}$ | −0.021 | $1.4 \times 10^{-8}$ | LBX1 |
| cg03839709 | 13 | 96743492 | −0.014 | $2.7 \times 10^{-4}$ | −0.014 | $3.4 \times 10^{-5}$ | −0.014 | $1.8 \times 10^{-8}$ | HS6ST3 |
| cg16419235 | 8 | 57360613 | −0.036 | $1.9 \times 10^{-4}$ | −0.029 | $8.3 \times 10^{-5}$ | −0.032 | $3.1 \times 10^{-8}$ | PENK |
| cg04317962 | 16 | 79623625 | −0.017 | $1.4 \times 10^{-6}$ | −0.019 | $2.9 \times 10^{-3}$ | −0.018 | $3.1 \times 10^{-8}$ | MAF |
| cg01429039 | 4 | 52918065 | −0.023 | $1.8 \times 10^{-4}$ | −0.020 | $1.1 \times 10^{-4}$ | −0.021 | $3.5 \times 10^{-8}$ | SPATA18 |
| cg03885399 | 1 | 47691550 | −0.020 | $4.4 \times 10^{-6}$ | −0.014 | $3.6 \times 10^{-3}$ | −0.017 | $4.7 \times 10^{-8}$ | TAL1 |
| cg04738965 | 3 | 147127662 | −0.037 | $4.0 \times 10^{-5}$ | −0.028 | $7.1 \times 10^{-4}$ | −0.033 | $6.2 \times 10^{-8}$ | ZIC1 |
| cg10954182 | 12 | 104532377 | −0.016 | $1.9 \times 10^{-4}$ | −0.009 | $2.1 \times 10^{-4}$ | −0.013 | $7.8 \times 10^{-8}$ | NFYB |
| cg10964367 | 8 | 1771973 | −0.025 | $1.3 \times 10^{-4}$ | −0.019 | $3.8 \times 10^{-4}$ | −0.022 | $9.4 \times 10^{-8}$ | ARHGEF10 |
| cg12739454 | 1 | 164290833 | −0.018 | $2.4 \times 10^{-4}$ | −0.012 | $2.4 \times 10^{-4}$ | −0.015 | $1.1 \times 10^{-7}$ | - |

*Δ Beta represents the difference in DNA methylation in cases as compared to controls (Beta Control-Beta PDB). Position in base pairs in reference to human genome build 37 (GRCh37). Chr, chromosome; CpG, cytosine-phosphate-guanine. All p-values are genome-wide significant based on Bonferroni corrected p-value < 0.05.

## Diagnostic capacity of differentially methylated markers

In order to determine whether differentially methylated markers might be of diagnostic value, we performed orthogonal partial least squares-discriminant analysis (OPLS-DA) in the discovery and cross-validation cohorts (refer to Materials and methods). The results are summarized in *Figure 4*. The OPLS-DA procedure was first performed using the combined set of significant DMS and DMR identified from the discovery set (*Pooled sites*; n = 2847, refer to Materials and methods for further details) and when the classifier was tested on the cross-validation set, it yielded an area under curve (AUC) of 92.8%. To identify sites with the highest predictive ability, we applied the net regularization extension of the generalized linear model approach on the *Pooled* sites, which resulted in the identification of 95 sites (which we also refer to as '*Best subset*' sites; *Supplementary file 5*), of the 2847 initial *Pooled* sites, as best discriminatory of PDB cases and controls (*Figure 1*). The OPLS-DA procedure performed on this *Best subset* resulted in an AUC of 82.5%. A rather superior performance in comparison to similarly trained classifiers based on the DMS (AUC = 67%), islands DMR (AUC = 76%), or promoter DMR (AUC = 79%) analyses. On the other hand, the AUC from a classifier restricted to the DMR gene bodies was 92%, which is similar to that obtained from the whole *Pooled sites* (AUC = 92.8, *Figure 3*).

Functional enrichment analysis of the 95 *Best subset* sites was consistent between Ingenuity Pathway Analysis (IPA) and Gene Ontology (GO) with many genes annotated to the following broad functional terms: *immune function, bone lesions and bone homeostasis,* and *viral processes*. Several identified genes fell into more than one category. Overlaying the IPA knowledge-based repository of molecular interactions identified a handful of functional links between the genes located in the *Best subset* sites, highlighting important functional subnetworks (*Figure 5A*). Additionally, we found that the effect size (absolute difference in DNA methylation between controls and PDB cases) was significantly higher for sites from the *Best subset* (mean ± SD; 0.011 ± 0.019) compared to the rest of those in the *Pooled sites* (0.007 ± 0.01; p-value=$1.9 \times 10^{-3}$). The magnitude of effect from each site in the *Best subset*, as calculated by the elastic-net regularization extension of the generalized linear model, is color-coded in *Figure 5B*.

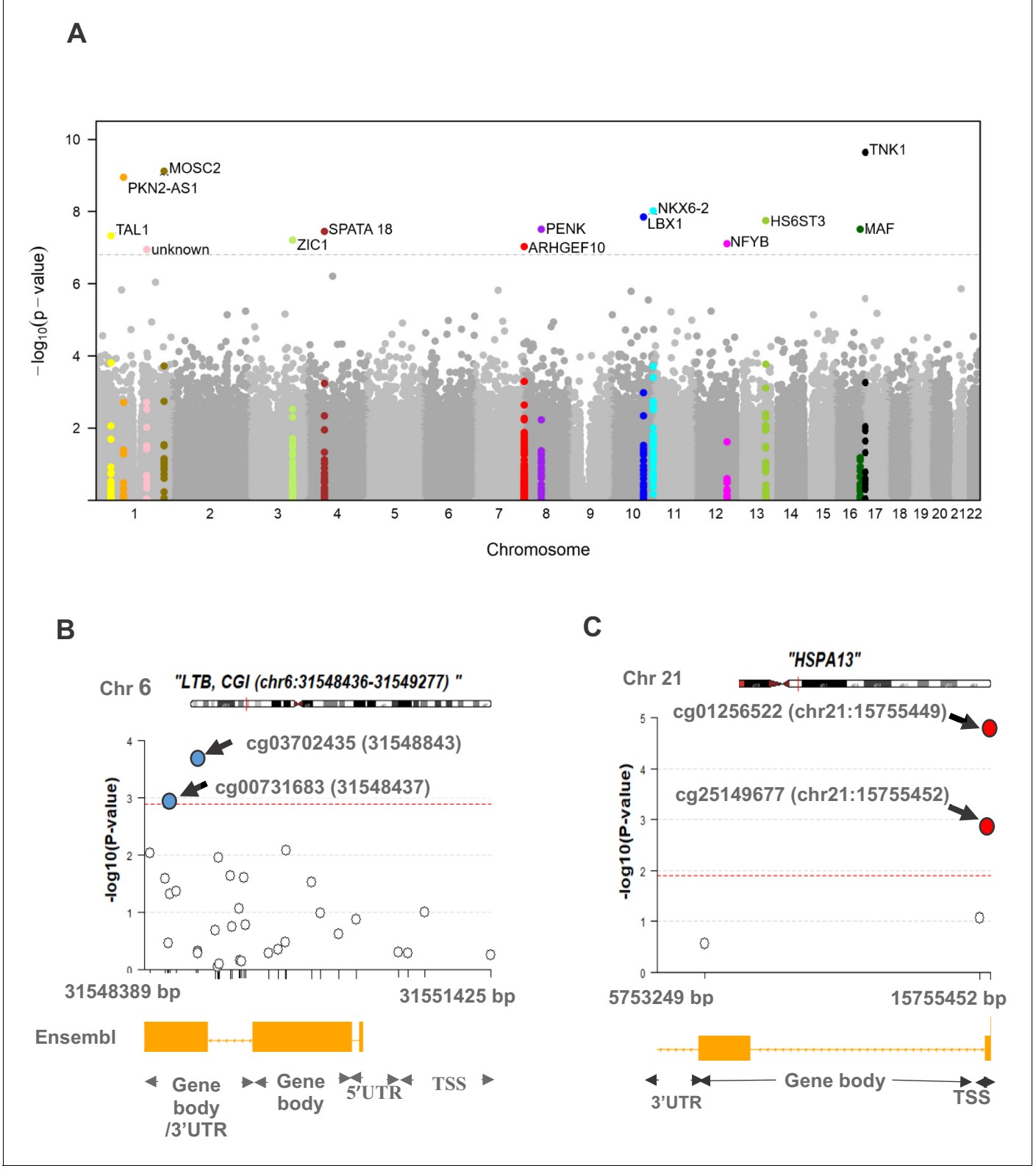

**Figure 2.** Differential methylation analysis comparing controls to PDB patients (n = 246). (**A**) Site analysis, a Manhattan plot showing the chromosomal positions (x-axis) versus the −log10 (p) of significant DMS and adjacent sites. For the Bonferroni significant sites however, the meta-analysis p-values are shown instead and highlighted in color. The horizontal dashed line indicates the Bonferroni corrected significance threshold (p<1.17×10⁻⁷). (**B, C**) Region analysis, showing the multitude of significantly hyper-methylated (red) and hypo-methylated (blue) sites from LTB (Bonferroni replicated from

*Figure 2 continued on next page*

*Figure 2 continued*

island analysis) and HSPA13 (Bonferroni replicated from gene body analysis). The dashed lines represent the FDR < 0.05 threshold for each region, which depends on the number of sites within the region (refer to Materials and methods).

The online version of this article includes the following figure supplement(s) for figure 2:

**Figure supplement 1.** QQ plots of expected versus observed –log10 p-values from site differential methylation analysis.

## Correlation of methylation profiles between blood and bone tissue

DNA methylation profiles are known to be tissue specific, and our DMS and DMR analyses were performed on blood, but the primary relevant tissue in PDB is bone. Therefore, we assessed if the methylation profiles for the DMS and DMR identified from this study are correlated between blood and bone tissue using previously published data by *Ebrahimi et al., 2021*. In their study, Ebrahimi et al. focused the correlation analysis on 64,349 CpG probes that fit their analysis criteria to define the most highly correlated positions, of which 28,549 CpG sites showed significant (FDR < 0.05) high correlation ($r^2 > 0.74$) between bone and blood. We assessed if CpG sites annotated to genes identified from our DMS and DMR analyses (*Tables 2* and *3*) showed high correlation between bone and blood as reported by *Ebrahimi et al., 2021*. Results showed that CpGs annotated to 8 of the 14 genes from our DMS analysis were among the highly correlated sites between blood and bone ($r^2 > 0.74$; FDR < 0.05; *Supplementary file 6*. For DMRs, of the 10 genes reported in our study (*Table 3*), 6 had at least one CpG with high correlation between blood and bone (*Supplementary file 6*).

## Expression quantitative trait-methylation (eQTM) analysis

eQTM analysis, based on the BIOS QTL (*Bonder et al., 2017*; *Bios QTL, 2021*), showed that the Bonferroni significant DMS cg10964367 was associated with the expression level of ARHGEF10 ($p=3.9\times10^{-9}$). Additionally, cg26724726 from gene body analysis was associated with the expression of LTB ($p=1.10\times10^{-5}$), and eight of the *Best subset* sites were associated with the expression of nearby genes (*Supplementary file 7*).

## Discussion

The present study is the first to investigate DNA methylation profiles in PDB. DNA methylation profiles from PDB patients were compared to controls, and meta-analysis of discovery and cross-validation revealed 14 genome-wide significant DMS. Many were located within or near genes with functional relevance to the pathogenesis of PDB including bone-related functions, such as osteoclast differentiation, or functions related to environmental triggers associated with PDB such as viral infection and mechanical loading. TNK1 is a tyrosine kinase that has a pivotal role in innate immune responses by regulating the Interferon-stimulated genes downstream of the JAK-STAT pathway

**Table 3.** Differentially methylated regions (DMR) in Paget's disease of bone.

| Region | Chr | Number of sites | Discovery p-value* | Cross-validation p-value* | Gene |
|---|---|---|---|---|---|
| Island | 6 | 53 | $1.40 \times 10^{-2}$ | $3.25 \times 10^{-4}$ | *LTB* |
| Island | 6 | 59 | $4.11 \times 10^{-3}$ | $2.47 \times 10^{-3}$ | *SKIV2L;RDBP* |
| Island | 10 | 49 | $2.65 \times 10^{-3}$ | $4.72 \times 10^{-3}$ | *EBF3* |
| Island | 11 | 49 | $3.57 \times 10^{-3}$ | $9.52 \times 10^{-3}$ | *CCND1* |
| Gene Body | 1 | 52 | $2.01 \times 10^{-5}$ | $3.14 \times 10^{-5}$ | *SDCCAG8* |
| Gene Body | 9 | 36 | $6.09 \times 10^{-3}$ | $1.20 \times 10^{-2}$ | *CACNA1B* |
| Gene Body | 8 | 51 | $2.49 \times 10^{-2}$ | $4.39 \times 10^{-3}$ | *RBPMS* |
| Gene Body | 21 | 5 | $3.19 \times 10^{-2}$ | $2.88 \times 10^{-3}$ | *HSPA13* |
| Gene Body | 2 | 52 | $3.80 \times 10^{-2}$ | $2.39 \times 10^{-3}$ | *PARD3B* |
| Gene Body | 22 | 34 | $4.49 \times 10^{-2}$ | $7.10 \times 10^{-3}$ | *BRD1* |

*P-values are adjusted for multiple testing using the Bonferroni method.

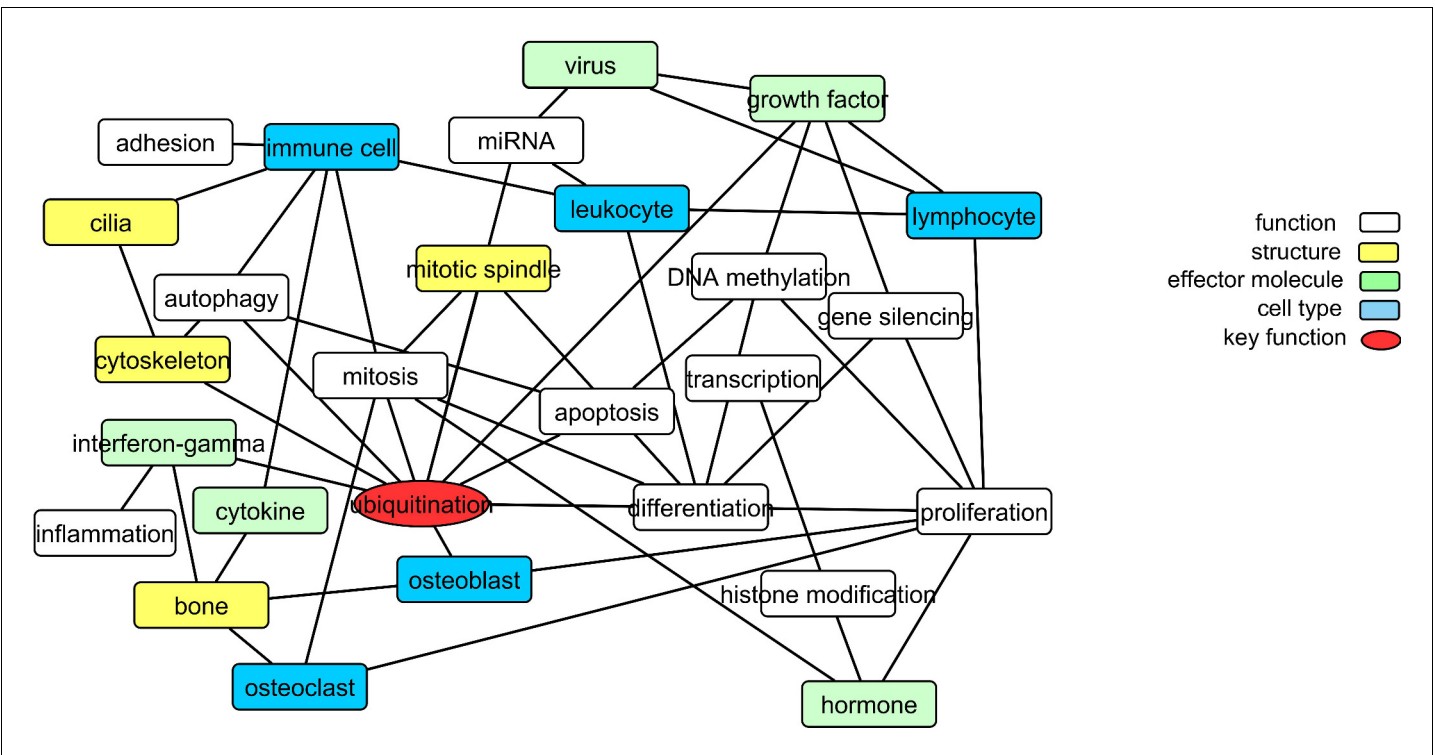

**Figure 3.** Translating the methylation data into functional networks. Nodes are functional, cellular, molecular, and sub-cellular keywords from GO annotations enriched amongst the *Pooled sites*. An edge between two nodes indicates that differentially methylated genes associated with the keyword in node one are significantly partially correlated with their counterparts from node 2 more often than can be accounted for by chance.

(*Ooi et al., 2014*). It has previously been associated with frontotemporal dementia (*Gijselinck et al., 2015*), which can co-exist with Paget's disease (*Watts et al., 2004*). MOSC2 is a member of the membrane-bound E3 ubiquitin ligase family that regulates endosome trafficking (*Zhang et al., 2018*). Less is known about the specific functions of transcription factors NKX6-2 and LBX1 in bone metabolism, but mutations in the latter are associated with Scoliosis. HS6ST3 plays a key role in the synthesis of heparan sulfate that potentiates key growth factors including the bone morphogenic protein BMP and Wnt (*Kuo et al., 2010*). PENK encodes for proenkephalin, the precursor of a range of effector molecules including pain-associated pentapeptide opioids as well as modulators of osteoblast differentiation (*Seitz et al., 2010*). Interestingly, PENK knockout mice have abnormal bone structure and mineralization (*Dickinson et al., 2016*). MAF was found to promote osteoblast differentiation, and heterozygous deletion of MAF in mice results in age-related bone loss associated with accelerated formation of fatty marrow (*Nishikawa et al., 2010*). SPATA18 is expressed in a variety of cancers including osteosarcoma, and its transcription is induced by p53 (*Bornstein et al., 2011*). TAL1 has been found to regulate osteoclast differentiation through suppression of their fusion mediator DCSTAMP (*Courtial et al., 2012*). The zinc finger protein ZIC1 has a role in shear flow mechanotransduction in osteocytes (*Kalogeropoulos et al., 2010*). Expression of ZIC1 in human was found to be increased in loaded compared to unloaded bone, and the increased expression in loaded bone is associated with reduced methylation in several CpGs in ZIC1 (*Varanasi et al., 2010*). NFYB confers chromatin access to other transcriptional regulators and is known to be involved in transition through cell cycle (*Ly et al., 2013*). Finally, the centrosomal ARHGEF10 has a role in the formation of mitotic spindle during mitosis (*Shibata et al., 2019*).

Our analysis was extended to identify regions with frequent but independent methylation changes in PDB amongst sites that are adjacent to each other. Genomic regions have traditionally been evaluated in epigenetics studies based on linear combinations of methylation data from residing sites or through meta-analysis of effects/p-values from an initial site-level differential methylation analysis. The novel approach presented in this study differs from the traditional methods in that enrichment of a region does not stem from frequent occurrences of correlated DMS within the

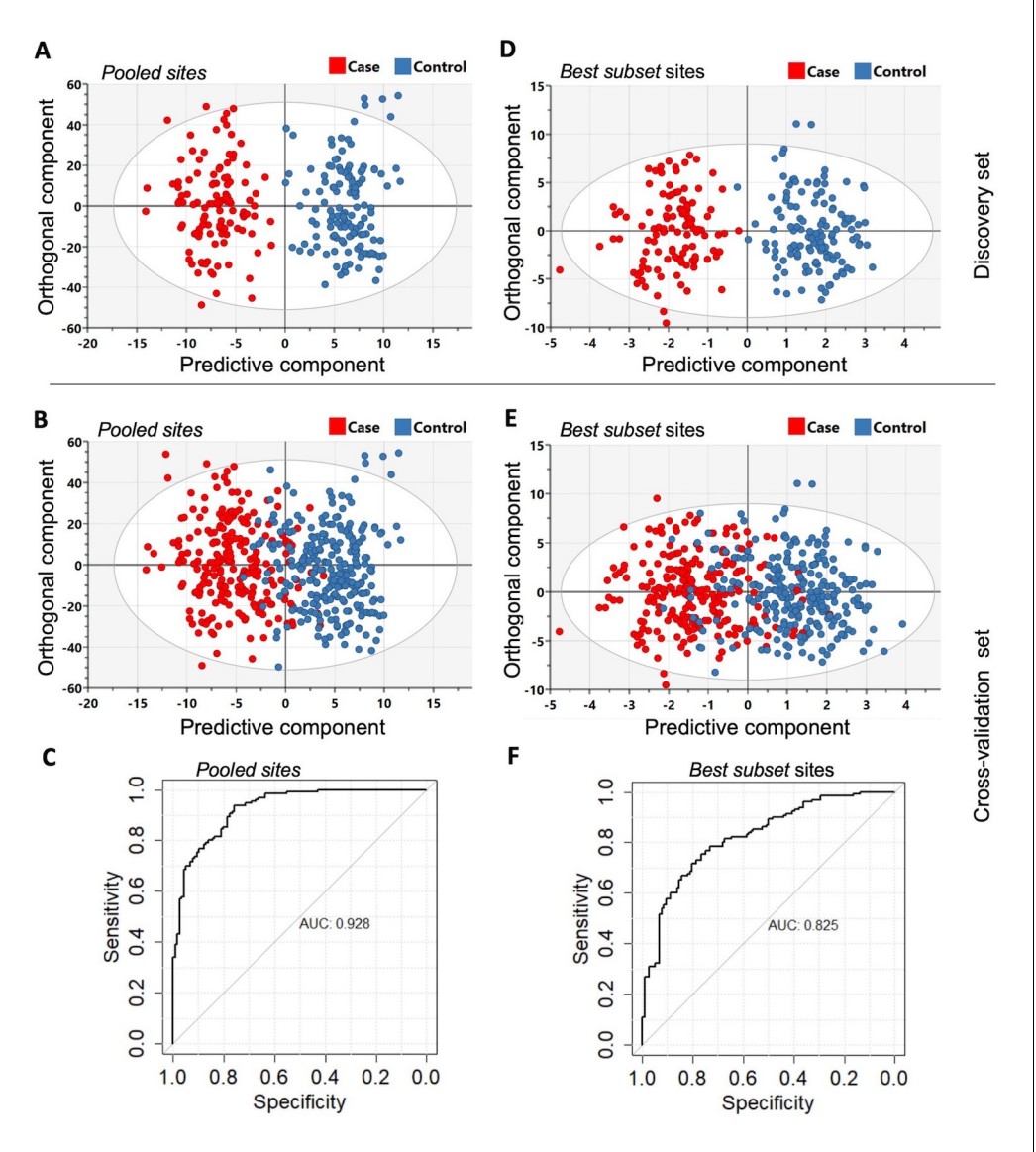

**Figure 4.** The orthogonal partial least squares-discriminant analysis (OPLS-DA) was performed using the *Pooled sites* identified from the discovery set (n = 246). (**A**) Classifier trained on all 2847 pooled sites with FDR < 0.05 (*Pooled sites*) from the discovery set. (**B**) Testing the classifier on the replication (or cross-validation) set. (**C**) ROC curve analysis yielded an overall sensitivity of 0.84, specificity of 0.81, and AUC of 0.928. (**D**) Classifier trained on the *Best subset* sites from Glmnet analysis (n = 95) using the discovery set. (**E**) Testing the classifier on the replication (or cross-validation) set. (**F**) ROC curve analysis showed an overall sensitivity of 0.77, specificity of 0.74, and AUC of 0.825. The Scatter plots show the predictive component that discriminates PDB cases from controls (x-axis) versus the orthogonal component representing a multivariate confounding effect that is independent of PDB (y-axis).

region but rather the accumulation of independent effects from residing sites. In other words, regions with the most, but unique site-level effects are prioritized. By doing so, our approach is advantageous in two ways: First, it allows for sites to be hyper- or hypo-methylated along the same region unlike the linear combination approach where opposing effects could neutralize one another. Second, it draws strength from the collective effects of neighboring sites whilst avoiding the redundancy of information from site-level analysis.

Four Bonferroni significant DMR were identified in islands, which were located near the following genes: LTB, a cytokine shown to stimulate osteoclast activity; SKIV2L, with an RNA helicase activity,

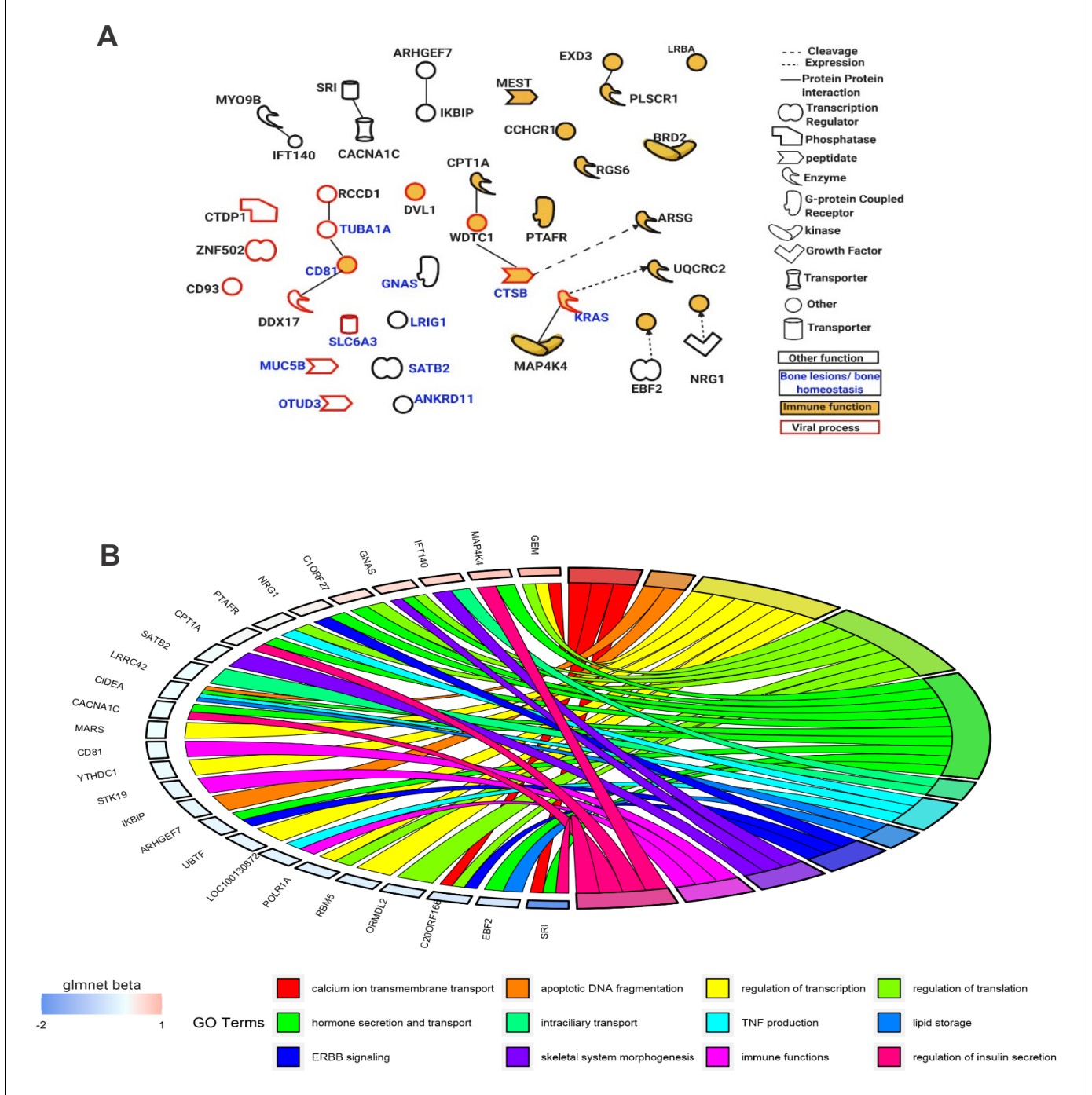

**Figure 5.** Functions of genes mapped near the *Best subset* of differentially methylated sites identified through the elastic-net regularization extension of the generalized linear model. (A) An IPA-based network showing a subset of these genes with functional interactions (edges) or mapping to one of three functional classes: immune, viral, and bone homeostasis. (B) An overview of GO biological processes significantly enriched amongst the Best subset together with their beta values from the Glmnet R package implementing the extended generalized linear model in question.

thought to be involved in blocking translation of viral mRNA and has been implicated in regulating host responses to viral infections (*Eckard et al., 2014*); EBF3, which is involved in bone development and B cell differentiation (*Seike et al., 2018*); and CCND1, a Wnt target that was reported to be upregulated in response to mechanical loading of bone (*Holguin et al., 2016*).

Additionally, six Bonferroni significant DMR in gene bodies were identified. These were located within genes with functions related to mitosis and ciliogenesis (SDCCAG8) (*Insolera et al., 2014*): TGFB1-mediated signaling (RBPMS) (*Shanmugaapriya et al., 2016*); calcium signaling (CACNA1B) (*Blair et al., 2007*); protein ubiquitination (HSPA13) (*Kaye et al., 2000*); cytoskeletal organization (PARD3B) (*Kohjima et al., 2002*); and histone acetylation (BRD1) (*Mishima et al., 2014*).

The *Pooled* sites identified from the discovery set were able to discriminate cases and controls with a considerable accuracy when tested on the cross-validation set. The Best subset analysis allowed the identification of a smaller subset of sites trading off the classification accuracy with the number of explanatory sites. The AUC of 82.5%, based on the 95 discriminatory sites from the best subset analysis, is promising, and future experiments are warranted to study its clinical applicability.

In terms of disease pathology, the DNA methylation data reflected many environmental triggers thought to be involved in PDB. Some of the genes amongst the DMS and the 95 *Best* subset were associated with immune antiviral responses (*Figure 5*, *Supplementary file 5*). This is of interest since a previous study in the PRISM cohort showed that levels of antibodies to Mumps virus were significantly higher in PDB cases compared to controls (*Visconti et al., 2017*). Although we and others have failed to detect evidence of ongoing virus infection in PDB, the above data is consistent with the hypothesis that host immune responses to infection may be altered in PDB.

Differential methylation of ZIC1 and CCND1 indicates possible differences between cases and controls in these genes, which are involved in mechanotransduction, a process that has been implicated in localization of bone lesions in PDB (*Gasper, 1979*). Our study also highlighted genes that regulate the cell cycle, vesicular transport, and cytoskeletal reorganization as being potentially involved in PDB. Other genes were identified that play a role in immune cell function, and these were strongly represented in the best subset of differentially methylated sites. This lends support to the hypothesis that PDB may be a disorder with an osteoimmunological basis (*Numan et al., 2015*) and should prompt further work to investigate host–environment interactions including studies of the microbiome in this complex but fascinating disease (*Ohlsson and Sjögren, 2018*).

Apart from providing new insights into the potential links between genes and environment in regulating susceptibility to PDB, this study has revealed the potential role of methylation signals as a biomarker for disease susceptibility. Potent bisphosphonates such as zoledronic acid can return the abnormalities of bone remodeling to normal in a large proportion of patients with PDB (*Reid et al., 2011*; *Reid et al., 2005*; *Tan et al., 2017*). Unfortunately, PDB often remains clinically silent until it has reached an advanced stage by which point irreversible skeletal damage may already have occurred (*Gennari et al., 2019*). This study raises the possibility that epigenetic markers, possibly when combined with genetic profiling, would be worth exploring as means of assessing the risk of developing PDB in people with a family history of the disorder so that early intervention can be considered where clinically appropriate.

One limitation of the study is the fact that the identified methylation changes were not shown to occur in the osteoclasts, which are the cells of main interest in PDB pathogenesis. This is primarily justified by the difficulty to collect bone tissue from PDB patients in a similarly sized cohort. Nevertheless, on comparison of our Bonferroni significant DMS and DMR (*Tables 2* and *3*), with a published list of highly concordant CpG sites between blood and femur bone tissue collected during hip replacement surgery (*Ebrahimi et al., 2021*), we noted considerable overlap. We found that CpGs annotated to 8 of the 14 genes from our DMS analysis were among the highly correlated sites between blood and bone (*Supplementary file 6*). For DMRs, of the 10 genes reported in our study (*Table 3*), 6 had at least one CpG with high correlation between blood and bone (*Supplementary file 6*). However, showing an epigenetic signature to PDB in the blood adds to the increasing evidence in the literature pointing to the possibility of pathogenic immune processes lying at the heart of PDB. More importantly, a predictive epigenetic signature in a readily accessible tissue such as the blood has clinical implication, also considering the silent nature of PDB and the possibility of avoiding much of the adverse symptoms of the disease with early diagnosis. Moreover, one needs to consider that blood also contains progenitors of bone cells and that white blood cells share a similar ancestry with osteoclasts.

Although the split-sample approach was meant to allow for validation of the results for increased statistical rigor, our cross-validation dataset is not totally independent from its discovery counterpart in that similar sources of noise and counfounding effects are present in both. However, the total cohort was obtained from a large number of centers across the UK representing most major cities,

which adds to the validity of our overall results. Another limiting aspect of our study was drawing functional relevance of our DMS and DMR by reference to tissue specific eQTMs from the BIOS QTL database, which were originally derived from blood. Therefore, the effects of the differential methylation from our candidates DMS and DMR on gene expression under PDB remain to be investigated. Finally, it is possible that the observed methylation changes reported in this study exist as a consequence of the disease; therefore, further prospective studies assessing their true potential as predictor biomarkers are warranted. Such studies could revolve around recruiting individuals with a genetic predisposition and/or family history of PDB for which the level of methylation of our 95 best subset sites can be routinely assessed. Such epigenetic measurements can then be linked to future disease onset if any, in the presence of appropriate controls.

# Materials and methods

**Key resources table**

| Reagent type (species) or resource | Designation | Source or reference | Identifiers | Additional information |
|---|---|---|---|---|
| Other | Infinium Human Methylation450 BeadChip | Illumina, USA | | DNA Methylation array |
| Software, algorithm | RnBeads | R | | Version 1.10.8 |
| Software, algorithm | SIMCA | Umetrics, Sweden | | Version 15 |
| Software, algorithm | IPA | Qiagen, Germany | | |
| Software, algorithm | GGM | R | | Version 2.4 |
| Software, algorithm | topGO | R | | Version 2.4 |

## Study subjects

The DNA samples were derived from UK-based PDB patients and controls who took part in the PRISM trial (Paget's Disease: Randomized Trial of Intensive versus Symptomatic Management) (ISRCTN12989577) (*Tan et al., 2017*). The PRISM trial is a multi-center study in which participants were recruited from 27 different clinical centers across the United Kingdom. The epigenetic analysis was conducted in 253 cases with clinical and radiological evidence of PDB and 280 controls who were spouses of PDB cases (n = 135) or subjects who had been referred for investigation of osteoporosis but had normal bone density upon examination by dual-energy X-ray absorptiometry (n = 131). The cohort was randomly divided into a discovery and cross-validation set comprising of comparable numbers of cases and controls (*Figure 1*). According to the study by Tsai and Bell, a 10% difference in the mean of CpG methylation level between cases and controls at genome-wide significance level of $10^{-6}$ requires 112 individuals in each group to achieve 80% EWAS power (*Tsai and Bell, 2015*). On this basis, our discovery set comprising of 116 cases and 130 controls is adequately powered, and the results are further validated in an equally sized cross-validation set.

## DNA methylation profiling

Genomic DNA was extracted from peripheral blood using standard protocols. Bisulfite conversion was performed on 500 µg of DNA using Zymo EZ-96 DNA methylation Kit (RRID:SCR_008968, Zymo Research, USA). DNA methylation profiling was performed using the Illumina Infinium HumanMethylation 450K array (Illumina, USA) by following the manufacturer's protocol. The R package *RnBeads* version 1.10.8 (RRID:SCR_010958) was used for quality control (*Müller et al., 2019*). Samples with low methylated or unmethylated median intensity (<11.0) were excluded (n = 35), along with samples with sex mismatch between reported and predicted sex (n = 0). Probes with the following criteria were excluded: detection p-value > 0.05, cross-reactive probes, containing a SNP within 3 bp of nucleotide extension site, or those located on sex chromosomes. Additionally, 723 sites were further excluded from the dataset for previously established association with smoking (*Ambatipudi et al., 2016*). A total of 56,356 probes were excluded from the initial 485,512 leaving 429,156 CpGs for analysis (*Figure 1*). The final dataset used for analysis comprised of 232 PDB cases and 260 controls. The *Enmix* method (*Pidsley et al., 2013*) was used for background correction, whilst *SWAN* was

used to achieve between and within array normalization. For all downstream analysis, the M-values, derived using the formulae $\log_2((\text{methylated signal} +1)/(\text{unmethylated signal} +1))$, were used.

## Statistics

An overview of the analysis performed in this study is shown in *Figure 1*, in what follows we provide details of each analysis step:

### Differential methylation analysis of sites

In order to account for the heterogenous cellular composition of the measured samples, the counts of the following cell types CD14 monocytes, CD19 B-cells, CD4 T-cells, CD56 NK cells, CD8 T-cells, eosinophils, granulocytes, and neutrophils were estimated using the *Houseman* reference method (*Houseman et al., 2012*), part of the *RnBeads* pipeline. The reference methylome was obtained from previously published methylation data measured from sorted blood cells comprising 47 samples (*Reinius et al., 2012*). These reference samples were normalized together with our data to make sure that extrapolation of cell type information was unaffected by differences between the two datasets.

We performed surrogate variable analysis (SVA) that captures additional unknown sources of variation based on joint methylation patterns amongst the different sites that do not correlate with the disease. The top 10 significant SVA components were extracted from the data using the SVA functionality in *RnBeads* (*Müller et al., 2019*).

In all statistical models described below, the term *confounders* refers to the following covariates: age, sex, array, bisulfite conversion batch, array scan batch, blood cell composition from the *Houseman* method (*Houseman et al., 2012*), and the top 10 SVA components. The term *phenotype* denotes the control/PDB state of each sample. The term *region* is used to describe clusters of sites along the genome including CpG islands, gene bodies, and promoters. CpG islands were delineated in the illumina array manifest file as well as RnBeads annotation libraries. Gene bodies and promoters were manually assigned. More specifically, sites mapping to the transcription start site (TSS) according to the manifest were attributed to a promoter region, whilst those falling at the 5′ untranslated region or gene body were assigned to a gene body region.

A general linear model based on the limma moderated standard error was used to assess differentially methylated sites (DMS) between cases and controls using the model: *CpG site ~phenotype +* confounders. The model was first run on all sites in the discovery set and all DMS with a significant FDR (<0.05) in the discovery set were assessed in the cross-validation set. Meta-analysis looking at the combined effect from both discovery and cross-validation was performed on the totality of probes using the R package *Metafor* (RRID:SCR_003450) (*Wolfgang, 2010*). The Bonferroni adjusted genome-wide significance threshold of $p=1.17\times10^{-7}$ (0.05/429,156) was used to identify Bonferroni significant DMS based on the meta-analysis p-values.

### Differential methylation analysis of regions

DMR were analyzed using binomial regression, member of the family of the generalized linear models (equivalent to logistic regression), in two steps:

First, the parameters of the *null* model, excluding the sites, were estimated as follows:

$$phenotype \sim confounders \tag{1}$$

Next, all *n* sites within a given region (island/gene body/promoter) were incorporated into the model as follows:

$$phenotype \sim confounders + CpG\,site_1 + CpG\,site_2 + .... + CpG\,site_n \tag{2}$$

The difference in the deviance (equivalent to the residuals in the linear model) between the null model [1] and the full model [2] follows a $\chi^2$ distribution with n degrees of freedom. A p-value for the effect of the region given n sites was calculated accordingly. The analysis effectively tests for the significance of improvement in the model fit with the addition of the methylation data from the region of interest. The generalized linear model outlined above was run initially on the discovery set. The model was then repeated on the cross-validation set on regions that were significant in the

discovery set at FDR < 0.05. A similar approach was used to derive the Bonferroni significant regions. In other words, the Bonferroni adjustment of regions in the cross-validation was based on the subset of regions found Bonferroni significant in the discovery set. Visualization of the effect of individual sites from selected DMR was conducted using R package coMET (*Martin et al., 2015*).

## Consolidating the DMS and DMR

In the generalized linear model for region effect outlined in model formulae [2], the beta values from the individual sites are indicative of the sites' level of association with the phenotype. This is effectively similar to the general linear model used for site-level analysis but with the important discrepancy that each site is being assessed while accounting for possible contributions of neighboring sites to the global effect of the region. We therefore extracted all the beta values form the full model in [2] from all the DMR. We then applied FDR-based multiple testing correction on the p-values corresponding to these beta values from fitting the model in [2] for each selected DMR separately. Sites with FDR < 0.05 were pooled with the DMRs to create a unified list of significantly methylated sites or *Pooled sites* (*Figure 1*).

## Discriminant analysis

Discriminant analysis was performed to assess the ability of the *Pooled sites* to tell apart cases from controls. We also used the elastic-net regularization extension of the generalized linear model, provided by the R package *Glmnet* (RRID:SCR_015505) (*Friedman et al., 2010*), to identify the best subset of discriminatory sites (designated *Best subset*) of the list of *Pooled sites*. We trained an orthogonal projection to latent structure discriminant analysis (OPLS-DA) classifier (*Boccard and Rutledge, 2013*), implemented in the software SIMCA ver. 15 (RRID:SCR_014688, Umetrics, Sweden), on the discovery data from *Pooled* and *Best subset* sites separately. Each model was then tested on the cross-validation set, and its performance was further assessed based on the value from receiver operating characteristic curve analysis. The sensitivity and specificity measures of the test were estimated based on a classification threshold equal to the median of the predicted scores by the OPLS-DA classifier. The *Best subset* sites were analyzed further to reveal enrichment in biological functions. This was conducted using IPA (RRID:SCR_008653, Qiagen, Germany) as well as the GO R package *topGO* (RRID:SCR_014798) (*Alexa, 2020*) based on the Fisher's exact test statistics.

## Partial correlation analysis of *Pooled sites*

Correlations in methylation patterns between CpG sites hold valuable information about how different biological functions are linked together in PDB. To this end, partial correlations between the *Pooled sites* were derived using the R package ggm (*Giovanni Maria, 2006*). The ggm partial correlations, based on the pooled sites, were used for drawing associations between GO biological process terms found enriched in the same set as follows: First, the extensive GO functional annotations enriched amongst the genes associated with the *Pooled sites* were manually reduced to a manageable, yet representative, set of keywords: For instance, GO categories 'regulation of proliferation', 'positive regulation of proliferation', and 'negative regulation of proliferation' were all reduced to 'proliferation'. The Fisher's exact test statistics was then used to assess whether the *Pooled sites* associated with a given keyword were correlated (based on the ggms) with their counterparts from another functional keyword more often than can be accounted for by chance alone. More specifically, for any two GO terms, we considered the significantly differentially methylated sites from genes associated with either terms. We then tested for enrichments of pairs of sites with significant ggms out of all possible pairs of sites across the two terms. Likewise, Fisher's test p-values<0.05 after FDR multiple testing correction were used to create pairs of functionally related keywords. The software Cytoscape (RRID:SCR_003032) (*Shannon et al., 2003*) was used to visualize these associations.

## eQTM analysis

To assess the effect of DNA methylation at CpG sites on the expression of nearby genes, we used data from the BIOS QTL browser (*Bonder et al., 2017*; *Bios QTL, 2021*).

## Acknowledgements

We wish to thank the patients and controls from the different centers who agreed to participate in this study. We would like to thank members of the PRISM trial research group across all participating centers for making DNA samples and data available for this study. We thank the Wellcome Trust Clinical Research Facility at Edinburgh University for performing the DNA methylation profiling. The research leading to these results has received funding mainly from the European Research Council to OMEA under the European Union's Seventh Framework Program (FP7/2007-2013)/ ERC gran agreement n° 311723-GENEPAD. This project has received funding from the European Research Council (ERC) to SHR under the European Union's Horizon 2020 research and innovation program (grant agreement n° 787270-Paget-Advance). This project received funding from the Paget's Association to OMEA. The PRISM trial was supported by grants from the Arthritis Research Campaign (13627) and the Paget's Association.

## Additional information

### Competing interests

Stuart H Ralston: has received research funding from Amgen, Eli Lilly, Novartis, and Pfizer unrelated to the submitted work. The author has no other competing interests to declare. The other authors declare that no competing interests exist.

### Funding

| Funder | Grant reference number | Author |
|---|---|---|
| European Research Council | FP7/2007-2013 (311723-GENEPAD) | Omar ME Albagha |
| European Research Council | Horizon 2020 (787270-Paget-Advance) | Stuart H Ralston |
| Paget's Association | | Omar ME Albagha |

The funders had no role in study design, data collection and interpretation, or the decision to submit the work for publication.

### Author contributions

Ilhame Diboun, Formal analysis, Writing - original draft, Writing - review and editing; Sachin Wani, genotyping; Stuart H Ralston, Resources, Data curation, Writing - review and editing; Omar ME Albagha, Conceptualization, Formal analysis, Supervision, Investigation, Writing - review and editing

### Author ORCIDs

Omar ME Albagha  https://orcid.org/0000-0001-5916-5983

### Ethics

Human subjects: The study was approved by the UK Multicenter Research Ethics Committee for Scotland (MREC01/0/53) and NHS Lothian, Edinburgh (08/S1104/8) ethics review committees. All participants provided written informed consent.

### Decision letter and Author response

Decision letter https://doi.org/10.7554/eLife.65715.sa1
Author response https://doi.org/10.7554/eLife.65715.sa2

## Additional files

### Supplementary files

• Supplementary file 1. List of replicated differentially methylated sites with FDR < 0.05.

- Supplementary file 2. List of replicated DMR at islands with FDR < 0.05.

- Supplementary file 3. List of replicated DMR at gene bodies with FDR < 0.05.

- Supplementary file 4. List of replicated DMR at promoters with FDR < 0.05.

- Supplementary file 5. List of Best subset sites from Glmnet analysis.

- Supplementary file 6. List of CpG, reported as correlated between bone and blood in Ebrahimi et al. (PMID: 32692944), mapping to the same genes as our significant DMS and DMR in Pagets disease.

- Supplementary file 7. List of expression quantitative trait-methylation (eQTM) sites from the Pooled sites. Highlighted in bold are the 8CpGs belonging to the best subset of sites (a subset of sites best explanatory of PDB).

- Transparent reporting form

## Data availability

Raw and processed methylation data generated in this study can be found at GEO under the accession GSE163970.

The following dataset was generated:

| Author(s) | Year | Dataset title | Dataset URL | Database and Identifier |
|---|---|---|---|---|
| Albagha OM, Diboun I, Ralston SH, Wani S | 2020 | Epigenetic analysis of Paget's disease of bone identifies differentially methylated loci that predict disease status | https://www.ncbi.nlm.nih.gov/geo/query/acc.cgi?acc=GSE163970 | NCBI Gene Expression Omnibus, GSE163970 |

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
