## [Decision Letter]

**Acceptance summary:**

There is substantial interest in finding circulating biomarkers for Paget’s disease of bone for diagnostic applications. DNA methylation patterns in peripheral blood mononuclear cells are identified in this study that are able to differentiate PDB cases from controls with a high level of accuracy. These candidate methylation sites and regions are associated with osteological and immunologic processes, suggesting functional relevance and may be of future clinical use.

**Decision letter after peer review:**

Thank you for submitting your article "Epigenetic analysis of Paget's disease of bone identifies differentially methylated loci that predict disease status" for consideration by *eLife*. Your article has been reviewed by 3 peer reviewers, and the evaluation has been overseen by a Reviewing Editor and Mone Zaidi as the Senior Editor. The reviewers have opted to remain anonymous.

Essential revisions:

1. As the authors point out in their discussion the genes identified in cells in the blood. The authors are correct that cells of the osteoclast lineage are in the blood; however, their study would be strengthened if their results could be verified in osteoclasts where possible. The reviewers all recognize bone samples are extremely challenging to obtain. Ebrahimi et al. (PMID: 32692944) provides a nice summary of correlation between methylation levels in blood and bone. The authors should consider looking whether their methylation candidates are correlated with methylation patterns in bone to augment their functional validation methods.

2. Given that the PDB cases were slightly older than controls in the discovery set, is this a concern for conclusions of this paper, given that incidence increases with age. Please comment on this issues. The cases and controls appear to be matched given that controls are spouses. This suggests 1:1 matching. Have the authors considered conditional models or other modeling approaches that more appropriately account for matching?

3. Regarding "The number of patients with SQSTM1 mutations" (p. 5): this supposedly refers to the cases (it's unclear whether the controls were tested for SQSTM1?)

4. Does OPTN have an eQTM in its vicinity?

5. Genomic/test statistic inflation is known to be a potential issue in EWAS. We encourage the authors to provide their q-q plots as a supplement to their analysis.

6. The authors used annotation information to define their regions. This approach depends on mapping information that is dynamic and somewhat subjective. Have the authors considered a more agnostic, data driven approach to identifying DMRs? Comb-p and other DMR approaches allow for identification of DMRs based on spatial correlation.

7. More information is needed about the model selection for the DMR analysis. Why chose a generalized binomial model? What link function did you use? Given case-control study design, readers are going to expect a logistic regression (logit link function). I encourage authors to provide brief rationale for choosing this model over more familiar options.

8. Probes are expected to be correlated within a DMR. I expect multi-collinearity to be an issue when modelling groups of correlated probes together as was done in the DMR analysis. Would authors be better suited modelling methylation on the outcome side of the model, fitting a linear mixed model? This model choice would fit better with the single probe analytic strategy.

9. Did the authors use the same Bonferroni adjusted p value in the discovery and validation sets? Was it based on the total number of probes tested in the discovery set or based on the number of probes carried forward for testing in the validation set? Both are reasonable approaches. However, additional clarification is needed.

10. The authors report using 10 components from the SVA model in their analysis. Can authors provide justification for 10 components, which seems high compared to similar studies.

11. The split sample cross validation approach was appreciated for its ability to maximize experimental rigor. However, this approach is distinct from a true external replication. Given that the 'training' and the 'test' sets come from the same overall population, we expect the 'replication' results to be optimistic relative to results from a true, external replication population. Given the absence of a suitable external replication population due the unique nature of the disease, this limitation is acceptable. Please discuss the potential limitations of this approach in the Discussion section. It is encouraged that the authors to refer to the 'replication' set as a 'cross-validation' set to more appropriately convey their experimental approach to the broader scientific community.

12. It is unclear how the partial correlations, using ggm, were used in the analysis. Please clarify.

---

## [Author Response]

Essential revisions:1. As the authors point out in their discussion the genes identified in cells in the blood. The authors are correct that cells of the osteoclast lineage are in the blood; however, their study would be strengthened if their results could be verified in osteoclasts where possible. The reviewers all recognize bone samples are extremely challenging to obtain. Ebrahimi et al. (PMID: 32692944) provides a nice summary of correlation between methylation levels in blood and bone. The authors should consider looking whether their methylation candidates are correlated with methylation patterns in bone to augment their functional validation methods.

We thank the reviewers and the editors for this thoughtful comment. Ebrahimi et al. (EPIGENETICS; 2021, 16(1): 92–105) investigated correlation in methylation profiles between blood and bone tissue in 12 subjects using Illumina MethylationEPIC BeadChip array. Bone samples were taken from the exposed proximal femur after removal of the femoral head from osteoarthritis patients. After quality control, Ebrahimi et al. focused the correlation analysis on 64,349 probes that fit their analysis criteria (to define the most highly correlated positions), of which 30,607 sites showed significant (FDR < 0.05) high correlation (r^2^ > 0.74) between bone and blood. Additional filter was applied to these sites to include those with at least 80% similar methylation profile between bone and blood (n = 28,549) which were reported as supplementary table in their paper.

We assessed if CpG sites annotated to genes identified from our DMS and DMR analyses (Table 2 and 3) showed high correlation between bone and blood as reported by Ebrahimi et al. Results showed that CpGs annotated to 8 out of the 14 genes from our DMS analysis were among the highly correlated sites between blood and bone (r^2^ > 0.74; FDR <0.05; Supplementary File 6). For DMRs, out of the 10 genes reported in our study (Table 3), 6 had at least one CpG with high correlation between blood and bone (Supplementary File 6). It is important to note that, in the study by Ebrahimi et al., only 64,349 CpG sites were tested for correlation, owing to the stringent criteria adopted by the authors to identify the list of highly concordant sites. Therefore, our DMS/DMR sites that did not feature in the list are not necessarily uncorrelated. Unfortunately, these sites cannot be investigated further since Ebrahimi et al. did not make their entire dataset available in public domain.

To address this point, A table has been added to the manuscript (Supplementary File 6) listing the sites with high correlation and the text has been modified to include and discuss these results.

2. Given that the PDB cases were slightly older than controls in the discovery set, is this a concern for conclusions of this paper, given that incidence increases with age. Please comment on this issues. The cases and controls appear to be matched given that controls are spouses. This suggests 1:1 matching. Have the authors considered conditional models or other modeling approaches that more appropriately account for matching?

The age factor was incorporated in the regression model in all analyses as indicated in the methods section. Additionally, the slight difference in age was only observed in the discovery set. There was no difference in age in the replication set (now termed cross-validation set in the revised manuscript). Given that we have adjusted for age in all analyses and that the reported loci were replicated in the cross-validation set, we don’t believe that our results are affected by the slight bias in age between the two groups in the discovery set. As for capturing the spouse information into an appropriate statistical model, spouses were only available for half of the PDB patients, therefore, a matched association analysis could only be fit on fewer subjects which could have compromised the power of the study.

3. Regarding "The number of patients with SQSTM1 mutations" (p. 5): this supposedly refers to the cases (it's unclear whether the controls were tested for SQSTM1?)

Controls were tested and they were all negative for SQSTM1 mutations as indicated in Table 1 which shows the number as “0”. We have clarified this in the revised manuscript.

4. Does OPTN have an eQTM in its vicinity?

OPTN did not feature amongst our list of significant DMS and DMR and there is no eQTM for OPTN in the BIOS QTL database. This suggests that the previously reported association between OPTN and PDB is mostly driven by genetic regulatory mechanisms as opposed to epigenetic regulation. In fact, the top associated SNP (rs1561570) from GWAS studies of PDB (Albagha et al., Nat Genet 2010) has been shown to significantly influence OPTN gene expression (eQTL) in osteoclast precursors as shown by Obaid et al. (Obaid et al., Cell Rep 2015).

5. Genomic/test statistic inflation is known to be a potential issue in EWAS. We encourage the authors to provide their q-q plots as a supplement to their analysis.

We have added the q-q plot to the manuscript as supplementary Figure 2-supplementary figure1. The genomic inflation factor for discovery set was 1.23 which is within the range reported by EWAS studies. For replication, we only analysed sites achieving FDR < 0.05 from discovery set. Guintivano et al. (EPIGENETICS 2020, 15(11):1163–1166) assessed genomic inflation in 16 published EWAS studies, of which 8 were EWAS for age and 8 were case-control studies for various conditions. Guintivano et al. found that the median inflation reported in case-control studies was 1.53.

6. The authors used annotation information to define their regions. This approach depends on mapping information that is dynamic and somewhat subjective. Have the authors considered a more agnostic, data driven approach to identifying DMRs? Comb-p and other DMR approaches allow for identification of DMRs based on spatial correlation.

We thank the reviewer for this comment. The Comb-P approach relies on correlation in p-values and applies a correction on these p-values to correct for multiple testing from neighbouring sites. This approach relies on summary statistics only (p-value) and cannot distinguish between sites with a redundant effect from those with an independent effect when it comes to explaining the phenotype. Also, Comb-P doesn’t take into account the location of CpG in relation to functional parts of genes (i.e. promoters, gene body, etc.) which is known to influence how methylation affect gene expression. The approach we used to define the DMR is widely used, and we believe that it is more powerful when individual-level data are available as it takes into account the above-mentioned factors. Lent et al. (BMC Genetics 2018, 19(Suppl 1):84, 28-31) compared various methods used to define DMR and found that Comb-P, DMRcate, and GlobalP detected similar DMRs. It would be useful to compare our approach to other methods, but we think this would be the focus of future methodology paper.

7. More information is needed about the model selection for the DMR analysis. Why chose a generalized binomial model? What link function did you use? Given case-control study design, readers are going to expect a logistic regression (logit link function). I encourage authors to provide brief rationale for choosing this model over more familiar options.

We apologize that our description of DMR analysis was not clear enough. The generalized linear model function for the binomial family in R is based on the logit link function and is therefore equivalent to the logistic regression. This has been clarified further in the method section.

8. Probes are expected to be correlated within a DMR. I expect multi-collinearity to be an issue when modelling groups of correlated probes together as was done in the DMR analysis. Would authors be better suited modelling methylation on the outcome side of the model, fitting a linear mixed model? This model choice would fit better with the single probe analytic strategy.

We agree with the reviewer that Multi-collinearity of probes within a DMR could occur and could cause the estimates of the model’s coefficients to be unreliable. However, our method was implemented in R in a way that it excludes regions with potential errors or warnings when fitting the model to make sure our estimates are correct. In general, this issue was limited and the proportions of regions that dropped out of the analysis was, on average, 5%. We thank the reviewer for suggesting new venues to be explored when modelling the region methylation effects, which are less susceptible to model fitting issues including multi-collinearity. We affirm the validity of their proposed approach. We are eager to explore these ideas to expand on the current methodology for region-level methylation analysis in future work.

9. Did the authors use the same Bonferroni adjusted p value in the discovery and validation sets? Was it based on the total number of probes tested in the discovery set or based on the number of probes carried forward for testing in the validation set? Both are reasonable approaches. However, additional clarification is needed.

For the site-level analysis, the Bonferroni adjustment was applied on the meta-analysis of the discovery and replication set, they were thus based on the entire set of probes (0.05/429,156 = 1.17 x 10^-7^) and this has been made clearer in the revised manuscript. As for the region level analysis, since the analysis does not lend itself to meta-analysis (for being based on a Chi-square distributed statistic), the Bonferroni adjustment of regions in the replication was based on the subset of regions found Bonferroni significant in the discovery. These additional details have now been added to the manuscript.

10. The authors report using 10 components from the SVA model in their analysis. Can authors provide justification for 10 components, which seems high compared to similar studies.

We chose to include 10 SVA components as a trade-off between correcting for as many unknown sources of variations as possible whilst avoiding issues of model overfitting. Using lower number of SVA resulted in higher genomic inflation. A previously reported EWAS study (Andrews et al., 2018, Molecular Autism 9, 40) applied a similar approach and used 19 SVA to achieve similar trade-off. However, their sample size (n=968) was larger than our study allowing the use of larger number of SVA.

11. The split sample cross validation approach was appreciated for its ability to maximize experimental rigor. However, this approach is distinct from a true external replication. Given that the 'training' and the 'test' sets come from the same overall population, we expect the 'replication' results to be optimistic relative to results from a true, external replication population. Given the absence of a suitable external replication population due the unique nature of the disease, this limitation is acceptable. Please discuss the potential limitations of this approach in the Discussion section. It is encouraged that the authors to refer to the 'replication' set as a 'cross-validation' set to more appropriately convey their experimental approach to the broader scientific community.

We have referred to the replication set as “cross-validation” as suggested by the reviewer. However, the study subjects were recruited from over 27 medical centres across the United Kingdom (UK) representing most major cities. We have also added text to discuss this point.

12. It is unclear how the partial correlations, using ggm, were used in the analysis. Please clarify.

The ggm partial correlations, based on the pooled sites, were used purely for drawing associations between GO biological process terms found enriched in the same set. We have incorporated this sentence in the method section on page 15 together with more details about how partial correlations between sites were used to infer relationships between associated GO terms. In particular, we added the sentence ‘for any two GO terms, we consider the significantly differentially methylated sites from genes associated with either terms. We then test for enrichments of pairs of sites with significant ggms out of all possible pairs of sites across the two terms’ to the same section in the revised manuscript.